# Clinical Application of Adenovirus (AdV): A Comprehensive Review

**DOI:** 10.3390/v16071094

**Published:** 2024-07-08

**Authors:** Md. Salauddin, Sukumar Saha, Md. Golzar Hossain, Kenji Okuda, Masaru Shimada

**Affiliations:** 1Department of Microbiology and Public Health, Faculty of Veterinary, Animal and Biomedical Sciences, Khulna Agricultural University, Khulna 9202, Bangladesh; salauddin.dvm@gmail.com; 2Department of Microbiology and Hygiene, Bangladesh Agricultural University, Mymensingh 2202, Bangladesh; sukumar.saha@bau.edu.bd (S.S.); mghossain@bau.edu.bd (M.G.H.); 3Department of Molecular Biodefense Research, Graduate School of Medicine, Yokohama City University, Yokohama 236-0004, Japan; okudanaika83@gmail.com

**Keywords:** adenovirus, vectors, gene therapy, cancer therapy, vaccine development

## Abstract

Adenoviruses are non-enveloped DNA viruses that cause a wide range of symptoms, from mild infections to life-threatening diseases in a broad range of hosts. Due to the unique characteristics of these viruses, they have also become a vehicle for gene-transfer and cancer therapeutic instruments. Adenovirus vectors can be used in gene therapy by modifying wild-type viruses to render them replication-defective. This makes it possible to swap out particular viral genes for segments that carry therapeutic genes and to employ the resultant vector as a means of delivering genes to specified tissues. In this review, we outline the progressive development of adenovirus vectors, exploring their characteristics, genetic modifications, and range of uses in clinical and preclinical settings. A significant emphasis is placed on their crucial role in advancing gene therapy, cancer therapy, immunotherapy, and the latest breakthroughs in vaccine development for various diseases.

## 1. Introduction

Adenoviruses belong to a prominent family of adenoviridae, non-enveloped viruses with a genetic makeup of double-stranded DNA that commonly result in respiratory illnesses in individuals of all ages [1]. Human adenoviruses consist of seven distinct species (A to G) and can also elicit a range of other health complications including gastrointestinal, ocular, urinary, and neurological disorders. Considering their prevalence, these viruses are commonly associated with the common cold, especially in young children. The testing of human samples yielded at least 116 ‘types’ of human adenoviruses, which consist of a much larger number of isolates (http://hadvwg.gmu.edu/ (accessed on 24 June 2024)) [2,3]. Scientists often use adenoviral vectors as potential vaccines for infectious diseases because they are easy to modify, have low pathogenicity, are simple to produce in the high-titer of the virus, are highly effective in delivering genes, and can carry large amounts of exogenous genetic material [4]. In the last 30 years, viral vectors like AdVs, adeno-associated viruses (AAVs), herpes simplex virus (HSV), lentivirus, and retrovirus have been extensively studied for their potential uses in gene therapy and vaccine development. Retroviruses integrate genetic material into infected cells, but their reliance on ongoing cell division presents a significant drawback when used as a viral vector. Lentiviral vectors, including human immunodeficiency virus (HIV), pose a higher risk of unpredictable genome integration in the host genome. Adeno-associated virus vectors can infect both dividing and non-dividing cells with a limited immune response, but they have size limitations in terms of cargo capacity (maximum 4.7 kb). In contrast, adenoviral vectors have distinct features such as they can deliver many copies of the genome into one cell, resulting in high gene expression; gene expression is temporary as the DNA is separate from the host genome; they can enter both dividing and non-dividing cells; and lastly, they provoke a strong immune response [5]. These qualities make adenoviral vector-based treatments mainly suitable for cancer therapy, single-gene disorders, and vaccination against infectious diseases such as human papillomavirus, hepatitis B virus, and hepatitis C virus [4,5]. Ever since adenovirus (AdV) was first discovered, it has proven to be a valuable tool for various therapeutic applications. Researchers have continuously refined AdV through various modifications such as fiber switching, incorporating ligands, and altering the structure of the fiber. These modifications, including the use of CRISPR-Cas9, have greatly enhanced the capabilities of AdVs as vectors and have also been adopted in other adeno-associated viruses [6]. In recent years, the adenovirus vector platform has captivated the scientific community as a potential tool in the fight against the SARS-CoV-2/COVID-19 pandemic [7]. Human adenoviruses, particularly type 5 (Ad5) and type 2 (Ad2), have been historically pivotal in fundamental virology studies. Notably, in 1962, it was observed that human Ad type 12 (Ad12) could induce tumors in newborn hamsters [8]. However, no conclusive evidence linking adenoviruses to tumor formation in humans has been established, and efforts to detect adenoviral DNA in human tumors have proved unsuccessful. A prevailing hypothesis, known as the ‘hit-and-run’ transformation model, suggests that adenoviruses may ignite cellular transformation with an initial ‘hit,’ followed by a ‘run’ [9]. The widespread occurrence of AdV infections in humans leads to the development of AdV immunity, which poses a significant obstacle to the effectiveness of HAdV vectors [10]. Similarly, the administration of an AdV vector provokes an anti-vectorimmune response, which can substantially hinder its effectiveness upon subsequent use. Considering that gene therapy often requires a substantially greater amount of vector compared to vaccines, delivering high doses of AdV vectors systemically could potentially cause toxicity in patients [11]. This review provides important insights into the applications, merits, and demerits of adenovirus as a vector and a chimeric and immunotherapeutic agent. 

## 2. Optimization of Adenovirus Vectors

The term optimization of adenovirus vectors involves refining their transduction efficiency and safety to enhance their effectiveness in delivering genetic material for gene therapy or other applications. Optimization of adenovirus vectors is vital for both experimental research and therapeutic applications [12]. Enhanced transduction efficiency ensures effective delivery of genetic material to target cells, while minimizing immunogenicity is crucial for avoiding immune responses. Optimization enables targeted tropism, directing vectors to specific cell types, and addresses safety concerns associated with toxicity [13]. It allows for the development of versatile adenovirus vectors that can accommodate different types of genetic payloads, such as genes, shRNAs, or CRISPR components [14]. A transposon-based system, employing Sleeping Beauty, optimized HEK293 cells for chimpanzee Adenovector (chAd) vector production, a Human Mastadenovirus C species. Swapping hAd5 E1 with chAd-C E1 via CRISPR/Cas9 revealed chAd-C E1 alone did not support HEK293 survival. To enhance chAd-C production, HEK293 cells were engineered to stably express ch.pTP, vital for chimpanzee Adenoviral DNA replication [15]. The introduction of exogenous ch.pTP expression has a substantial positive impact on the packaging and amplification of recombinant chAd-C vectors. Therefore, the modified HEK293ch.pTP cells may serve as a more effective cell line for producing these vectors [15]. The Ad5 system was optimized, enhancing capsid protein expression and promoting their assembly into virus-like particles (VLPs). The engineered Ad5[PVP2]OP vector, incorporating FMDV sequences under an optimized CMV promoter in reverse transcriptional orientation, exhibited a substantial (~14-fold) increase in protein expression compared to unmodified Ad5 vectors [16]. Regulatory compliance, scalability, long-term stability, and versatility are also achieved through optimization, facilitating the development of effective and safe gene therapies [17]. These efforts collectively contribute to advancing scientific research and improving the precision and efficacy of gene delivery for therapeutic purposes.

### 2.1. Replication Competent Adenovirus

A Replication Competent Adenovirus (RCA) is an adenovirus that retains the ability to replicate and reproduce within host cells. Adenoviruses are a family of viruses that can infect various tissues and are often used as vectors in gene therapy and vaccine development. Replication-competent adenoviruses differ from replication-deficient ones in that they are capable of undergoing multiple rounds of replication, producing new viral particles and potentially causing a spreading infection. Replication-competent (oncolytic) adenovirus vectors are employed for cancer gene therapy. Oncolytic vectors are engineered to replicate preferentially in cancer cells and to destroy cancer cells through the natural process of lytic virus replication. Many clinical trials indicate that replication-defective and replication-competent adenovirus vectors are safe and have therapeutic activity. However, the use of replication-competent adenoviruses poses safety concerns, as uncontrolled replication could lead to unintended side effects or adverse reactions. To address these concerns, researchers often design safety features, such as incorporating specific mutations to limit replication to certain cell types or including mechanisms for controlling and terminating viral replication. Careful consideration of the potential risks and benefits is crucial when working with replication-competent adenoviruses in a therapeutic context. 

A comprehensive examination of Replication Competent Adenoviruses (RCAs) is imperative for various reasons. Understanding the safety profile of RCAs is crucial for identifying potential risks in different cell types and tissues, ensuring safety in therapeutic applications. Research is needed to enhance the efficiency of RCAs in delivering therapeutic genes, optimizing their replication within specific cells while minimizing adverse effects. Developing RCAs with increased tissue specificity requires a detailed understanding of their interactions with different cell types, enabling the design of viruses that selectively replicate in desired locations [18]. Studying the immune response to RCAs is essential for assessing potential host reactions, guiding strategies to modulate immune responses and ensure therapeutic effectiveness. Preclinical studies, including testing in animal models, are essential to establish the safety and efficacy of RCAs before clinical translation. Further research is crucial for obtaining regulatory approval, providing comprehensive data on safety, efficacy, and mechanisms of action for RCA-based treatments [18,19]. 

Replication Competent Adenoviruses (RCAs) encounter challenges in gene therapy, including safety issues linked to uncontrolled replication and off-target effects. Immunogenicity and pre-existing immunity may limit their efficacy, and concerns about insertional mutagenesis raise integration risks [20]. The constrained payload capacity restricts the genes in a single vector. Ethical and regulatory scrutiny surrounds the use of live viruses, and the complex manufacturing process requires strict quality control. Addressing these challenges demands ongoing research to refine RCAs, enhance safety, and minimize risks, with comprehensive preclinical and clinical studies essential before widespread clinical use. Despite these challenges, RCAs are widely used in research and therapy for efficient gene delivery, transient gene expression, vaccine development, and biomedical research [21]. Their versatility as viral vectors and potential in emerging infectious disease vaccines highlight their advantages, emphasizing the ongoing need for safety refinement and design improvement across applications [22]. 

Given the scarcity of clinical safety data on residual RCA, studies on restrictively replication-competent or oncolytic adenoviruses (Conditionally Replicating Adenoviruses) provide valuable evidence. Oncolytic vectors are specifically designed to replicate in a controlled manner within cancer cells, leading to tumor lysis. An example of such a vector is ONYX-15, which contains an intact replication-permissive E1A region but lacks the p53-suppressive E1B region. This design theoretically allows replication only in cancer cells that lack functional p53 [23]. However, earlier constructs have been reported to lack specificity in replication [24], prompting further efforts to enhance the selectivity of oncolytic adenoviruses [25]. 

### 2.2. Replication Deficient Adenovirus (Human Ad5, hAd36, Chempanzee)

Engineered replication-defective adenoviruses (rAd) offer efficient expression of foreign inserts while minimizing adenovirus protein expression, making them ideal for vaccine and gene therapy applications.

Replication-deficient adenoviral (Ad) vaccines have played a significant role in addressing the SARS-CoV-2 pandemic. Vaccines utilizing three Ad types—HAdV-C5 (Ad5), HAdV-D26 (Ad26), and chimpanzee Y25 (ChAdOx1)—have received emergency use authorization (EUA) in the United States, European Union, South America, Asia, and Africa, and full approval in Russia [26]. 

RD Ad vectors, widely used in vaccines, elicit robust humoral and T cell responses, particularly favoring a T helper 1 type response to the vector-expressed transgene. Notably, the STEP study, employing a mixture of three Ad5-based RD vectors as an HIV vaccine, was conducted by Merck in around 3000 volunteers across multiple countries starting in 2004 [27]. Despite well-tolerated intramuscular injections of up to 10^11^ vp per injection, the trial was halted in 2007 due to an unexpected outcome: vaccinated individuals showed higher rates of HIV infection compared to the placebo group. This increase was observed in uncircumcised men-who-have-sex-with-men with pre-existing antibodies to Ad5, and the underlying cause remains unexplained [28].

In 2009, the NIH initiated the HVTN HIV vaccine clinical trial, employing a phase II approach. The trial involved a prime with a DNA-based vaccine expressing HIV proteins, followed by a boost with an RD Ad5 vector expressing the same proteins. However, the trial was stopped in 2013 as the vaccine regimen did not lower the rate of HIV infection or alleviate infection in volunteers who acquired HIV during the trial [29].

Unlike RD Ad5-vectored HIV vaccines, an RD Ad5-based tuberculosis vaccine (AdHu5Ag85A) shows promise [30]. In a phase I study involving 26 volunteers, the vaccine, expressing a key Mycobacterium tuberculosis antigen, was administered via a single intramuscular injection. Results indicated safety, tolerability, and immunogenicity in both Bacille Calmette-Guerin (BCG)-immunized and BCG-naïve groups, with boosted CD4^+^ and CD8^+^ T cell responses in the BCG-vaccinated group. Notably, preexisting Ad5 immunity did not impact vaccine safety or efficacy [31].

One another trial on HIV, no correlation was observed between T-cell responses and neutralizing antibodies (NAs) to Ad5. At time points where antibodies to Ad5 and T-cell responses to the recombinant gene products were measurable, there was no enhancement in the magnitude or activation state of Ad5-specific CD4^+^ T cells. These findings suggest that rAd5-based vaccines with deletions in the E1, E3, and E4 regions do not lead to significant expansion of vector-specific CD4^+^ T cells [32].

### 2.3. Modification Replication Deficient Ad

‘Modification replication-deficient adenovirus’ refers to an adenovirus that has been genetically altered to disable its ability to replicate efficiently within host cells. In the context of gene therapy or vaccine development, this modified adenovirus serves as a vector for delivering therapeutic genes or antigens into target cells. The term ‘replication-deficient’ indicates that the virus can no longer reproduce itself effectively, minimizing the risk of uncontrolled viral replication and enhancing the safety of the therapeutic approach. This design allows the adenovirus to deliver its payload without causing a widespread infection, making it a controlled and safer tool for various biomedical applications.

‘Replication-deficient adenovirus’ refers to an adenovirus that has undergone genetic modifications to render itself incapable of replicating efficiently within host cells [33,34]. Adenoviruses, known for causing respiratory, gastrointestinal, and various infections in humans, are frequently employed as vectors in gene therapy and vaccine development. The term ‘modification replication-deficient Ad’ signifies alterations made in the viral genome to achieve replication deficiency or impairment. Crucially, this modification ensures safety by preventing uncontrolled viral replication while still allowing the adenovirus to deliver its genetic payload into target cells, eliciting the desired therapeutic response. In gene therapy or vaccine applications, the replication-deficient adenovirus serves as a vector for carrying therapeutic genes or antigens into cells [35]. Although the virus can infect cells and deliver genetic material, its modified machinery prevents the production of more virus particles, minimizing the risk of uncontrolled replication and safeguarding the therapeutic approach’s safety. The specific modifications made to create a replication-deficient adenovirus typically involve disabling or deleting key viral genes essential for replication, aiming to balance efficient gene delivery with safety considerations [36].

### 2.4. Merit and Demerit of Adenovirus Type 5 

Ad5 has been extensively studied and used in various gene therapy and vaccine applications.

It has efficient gene-transfer capabilities and is well-characterized. Many people have pre-existing immunity to Ad5, which can be a limitation in some cases, but it may also be an advantage in others, depending on the context. Immunization involving a DNA vaccine prime followed by an adenovirus type 5 (Ad5) boost elicited a protective immune response against the SHIV challenge in monkeys. However, the hepatocellular tropism of Ad5 limits the safety of this viral vector [37]. Scientists tested a new HIV vaccine on mice and monkeys using a modified adenovirus. This adenovirus, called Ad5/35, had low liver toxicity in animals. The vaccine, containing the HIV clade C gag gene, prompted strong HIV-specific immune responses in both mice and monkeys [38].

A combination vaccination approach using genetic material (rDNA) for specific HIV components, Gag and Env, followed by booster shots of a modified adenovirus (rAd5/35) resulted in stronger immune responses compared to using the genetic material or the adenovirus alone. When tested with a harmful simian human immunodeficiency virus (SHIV), animals that received either the combination vaccine or just the adenovirus alone maintained higher levels of a crucial immune cell (CD4^+^ T cells) and effectively controlled the amount of virus in the bloodstream. These findings suggest that the rAd5/35-based vaccine shows promise for clinical use [39]. An earlier investigation revealed that healthy blood donors in developed regions commonly have neutralizing antibodies against Ad5, while such antibodies against Ad35 are uncommon. Additionally, individuals at risk of AIDS were found to exhibit a significantly higher presence of Ad5 antibodies compared to Ad35 antibodies [40]. 

One study focused on developing an HIV-1 vaccine using the widely-used Adenovirus type 5 vector (Ad5). They modified the Ad5 by adding a key component (ELDKWA) known for broadly neutralizing various HIV-1 strains. Additionally, they included the HIV-1IIIB envelope (Env) gene in the vaccine to stimulate specific immune responses. The modified Ad5, when given in high doses, effectively triggered immune responses, producing cells targeting HIV-1 and generating antibodies against the virus. In laboratory tests, the antibodies showed the ability to neutralize a broad range of HIV-1 strains. That research represents a promising step in creating an Ad5-based HIV-1 vaccine that induces both cellular immune responses and neutralizing antibodies against diverse HIV-1 strains simultaneously [41].

The effectiveness of Ad5 vectors in humans has been hindered because most adults have immune defenses (neutralizing antibodies) against Ad5 due to prior infections. To address this issue, we altered a specific part of Ad5 called the hexon epitope in its fifth hypervariable region. This modification allowed the Ad vector to evade the neutralizing antibodies against Ad5. Mice injected with this modified vector produced lower levels of individual neutralizing antibodies compared to those injected with the original vector. Notably, mice with pre-existing immunity against Ad, when immunized with the modified vector, showed much stronger cell-mediated responses against the virus compared to those immunized with the original vector [42].

The Ad5/F35 vectors exhibit expanded tropism, infecting CAR-negative cells through the CD46 receptor, unlike Ad5 vectors. With a larger cargo capacity (~8.2 kb), they accommodate diverse genetic material, suitable for various applications. These vectors, remaining as episomal DNA in the nucleus upon transduction, pose a low risk of host genome disruption. Achieving very high titers (>10^10^ IFU/mL) through re-infection of packaging cells sets them apart from lentivirus, MMLV retrovirus, or AAV. The vector proves effective in both in vitro and in vivo settings, providing versatility. Engineered for safety, it lacks essential virus production genes, ensuring replication incompetence except when transducing packaging cells [43,44].

Adenoviral vectors present certain characteristics that impact their application. Firstly, the vector’s DNA does not integrate into the host cell genome but persists as episomal DNA, particularly prone to loss in dividing cells [45]. Additionally, the live virus generated by adenoviral vectors triggers a robust immune response in animals, limiting its in vivo utility [46]. Furthermore, utilizing adenoviral vectors involves intricate processes, including live virus production in packaging cells and the measurement of viral titer, making it technically demanding and time-consuming [47]. Key elements in the vector include the 5′ inverted terminal repeat (ITR), 3′ ITR, Ψ packaging signal, promoters, Kozak consensus sequence, open reading frame (ORF), polyadenylation signals, and markers, each serving specific functions in facilitating gene expression and vector replication [48]. The vector design also incorporates the ΔAd5/F35 configuration, enhancing tropism for cells with low or lacking coxsackievirus and adenovirus receptor expression [49]. The plasmid backbone, containing an ampicillin resistance gene and a PacI restriction site, ensures efficient vector linearization for packaging [50]. 

Three generations of adenoviral vectors have been developed thus far. In order to create the first-generation of Ad vectors, transgenic cassettes up to 4.5 kb in length were engineered by replacing the E1A/E1B region [51]. The E2/E4 site was also deleted in the second-generation of adenoviral vectors, which increased the transgene capacity even further, although the overall production capacity remained low because of the decreased replication capacity of the producer cell lines [52]. The third-generation adenovirus vectors are commonly called helper-dependent or gutless adenovirus, where all the viral sequences are deleted with the exception of ITRs and the packaging signal. Despite the maintenance of high transduction and tissue tropism, the resulting in vivo immune response in these viral vectors is highly reduced compared to the first- and second-generation adenovirus vectors [53] (Figure 1 and Figure 2).

### 2.5. Merit and Demerit of Ad 11 and Ad 35

Research on cardiovascular gene therapy has concentrated on evaluating the safety and effectiveness of uncommon alternative serotypes and/or genetically modified adenoviral capsid protein-modified vectors after local or systemic administration. Numerous vectors, such as HAd-11, HAd-35, and HAd-20–42–42, have been found in pre-clinical research to be potential platforms for local and systemic targeting of smooth muscle and vascular endothelial cells [54].

Although this approach may allow luminal transgene distribution to damaged vasculature, there is a chance that it will have negative side effects. Due to the limited presence of CAR on vascular smooth muscle cells (SMCs) and endothelial cells (ECs), which increases the risk of toxicity and enhanced immune responses, substantial doses of HAd-5 are necessary for successful ex vivo saphenous vein graft (SVG) transduction.

Drug-mediated immunosuppression is one strategy to minimize immunogenicity and improve target cardiovascular cell transduction [44]. Ad11, or adenovirus type 11, possesses broad tropism, which makes it possible to infect a variety of cell types. This property is important for gene therapy applications.

Nevertheless, one disadvantage of this approach is the possibility of interference due to immunity that has developed following natural infections in the population [45]. In contrast, coxsackievirus and adenovirus receptors (CAR) are not the only receptors that Ad35 offers. It can also help overcome the problem of poor CAR expression in target cells [55]. Ad35 has a limited prevalence in the human population, which could be advantageous or disadvantageous depending on the context of use. However, it may also diminish pre-existing immunity. The significance of carefully designing and utilizing adenovirus vectors for gene therapy or vaccine production is highlighted by these factors [45,46].

### 2.6. Merit of Chimera Ad 5/35 

Adenovirus fiber proteins help the virus attach to cell membranes. Most Ad vectors use Ad type 5 (Ad5) and a receptor called CAR. Ad35, a different type, uses unknown receptors. Ad vectors with a mix of Ad5 and Ad35 fibers show broader cell attachment and can carry larger genetic material. This chimeric vector might enhance gene therapy, especially in cells with low CAR expression [43].

Ad5/35 refers to a chimeric adenovirus type 5 and type 35 vector. Adenoviruses are commonly used as vectors in gene therapy and vaccine development due to their ability to efficiently deliver genetic material into target cells. The chimeric Ad5/35 vector combines genetic elements from adenovirus type 5 (Ad5) and adenovirus type 35 (Ad35) [56]. 

Adenovirus type 35 has a different tropism compared to Adenovirus type 5, meaning it may infect different types of cells. Combining these types in a chimeric vector may broaden the range of cells that the vector can infect, potentially improving its effectiveness [44]. Ad5 is a common human adenovirus, and many people have pre-existing immunity to it due to previous exposure. By incorporating elements from Ad35, which is less prevalent in the human population, the Ad5/35 vector may be less affected by pre-existing immunity to Ad5.

The combination of different adenovirus types in a chimeric vector may enhance the immune response to the delivered antigen. This can be beneficial in the development of vaccines, where a robust and diverse immune response is desirable [57]. Chimeric adenovirus vectors provide researchers with a versatile platform for designing vaccines against various pathogens. The modular nature of adenovirus vectors allows for the insertion of genetic material encoding antigens from different pathogens.

### 2.7. Merit of Fiber Modified (Present Epitope) Ad

Modifying fiber proteins presents a promising approach to overcome the limitations of adenovirus (Ad) infection dependence on CAR. Two strategies have been employed: adding foreign peptides to the HI loop or C terminus of the fiber knob, and substituting fibers from different Ad serotypes that bind to receptors other than CAR. Both methods broaden or alter Ad tropism by allowing the modified fiber protein to bind with cellular receptors. This fiber modification, crucial for changing adenovirus tropism, impacts the virus’s ability to bind to its cellular receptor, with the RGD4C peptide family, particularly when incorporated into the fiber knob, demonstrating high-affinity binding to integrin receptors and promoting adenovirus infection. However, inserting RGD4C into the receptor binding site of the fiber knob, such as in the HI loop of HAdV-35 fiber knob, may disrupt the interaction between the fiber and its native receptor [58]. Fiber-modified Ad vectors, constructed using a simple in vitro ligation method, demonstrate improved gene transfer, overcoming limitations in cells lacking the primary receptor. Comparative studies with RGD and K7 peptides show that vectors with both peptides in the HI loop or C terminus display superior gene transfer and broader tropism, providing valuable insights for gene therapy and functional studies [59].

Recombinant adenovirus (Ad) vectors are extensively utilized in gene-transfer studies in vitro and in vivo, as well as in clinical gene therapy trials [60]. Overcoming challenges such as pre-existing immunity, liver tropism, and high toxicity associated with first-generation Ad vectors (FG-Ad), significant advancements have been made in developing genetically and chemically modified adenoviral vectors. These efforts have resulted in more sophisticated vectors for gene therapy, demonstrating an enhanced safety profile and improved transduction ability across various tissues [61].

The natural cellular entry mechanism of HAdV-C5 vectors involves the homotrimeric fiber-knob protein interacting with the coxsackie and adenovirus receptor (CAR), followed by the penton base RGD-motif interacting with αvβ3- or αvβ5-integrins on the cell membrane. As both CAR- and RGD-binding integrins are lacking in human T cells, these cells pose a challenge for transduction by HAdV-C5 without modifications to either the vector or the cell. In response, T cells have been engineered to express either CAR22 or αVβ3/5 integrins, rendering the cells susceptible to adenoviral infection [62].

Efforts to enhance adenovirus (Ad) gene-transfer efficiency include modifications to the fiber protein, such as inserting small peptides or replacing the Ad5 knob with the Ad35 knob, leading to improved CD46-dependent gene transfer [63]. Redirecting Ad5 tropism is crucial for precise gene delivery as the fiber plays a key role in Ad5 infection. The Ad fiber protein is a major determinant in mediating Ad binding to the coxsackievirus and Ad receptor (CAR). Despite unrelated origins, adenoviruses, alphaviruses, and poxviruses share a capacity for high-level expression, especially in vaccination and immunotherapy applications [64,65]. Notably, Ad type 37 (Ad37), associated with severe ocular infections, struggles to efficiently use CAR for host cell infection despite having a CAR binding site in its fiber knob.

## 3. Application of Adenovirus

Adenovirus vectors have proven to be a viable option for gene therapy, effectively delivering genetic material into targeted cells to address various health conditions. With their versatility in infecting different cell types, they have shown potential in cancer treatment and vaccine development. However, challenges remain, including immune reactions to the vector and the potential for genetic alterations. Moving forward, efforts are focused on optimizing vector design for improved safety and effectiveness, investigating alternative serotypes to evade immune detection, and utilizing advanced genetic engineering techniques to minimize any potential drawbacks. These advancements pave the way towards broader usage of this approach in clinical applications. 

Over time, adenovirus (first, second, and third-generation) vectors have transformed into powerful tools for treating diseases and advancing gene therapy. Not only can they facilitate gene expression, but they also play a crucial role in stimulating the body’s adaptive immune responses. Their potential for combating various illnesses in both humans and animals cannot be ignored, with successful treatments for Malaria, HIV, Zika virus, Nipah virus Ebola virus, Hepatitis B and C virus, Chikungunya virus, foot-and-mouth disease virus, rabbit hemorrhagic disease virus, feline immunodeficiency virus, Rift Valley fever virus, rabies virus, and PRRS already demonstrated. The ongoing progress of adenovirus vectors offers exciting possibilities for addressing a vast range of medical challenges through cutting-edge and precise therapeutic approaches [35].

The rapid development of SARS-CoV-2 vaccines has led to the approval of four adenovirus vector vaccines: ChAdOx1 nCoV-19 (VAXZEVRIA, AZD1222) by AstraZeneca and the University of Oxford in the UK, Sputnik V by the Gamaleya Research Institute in Russia, Ad26.COV2-S by Johnson & Johnson in Europe, and Ad5-nCOV by CanSino Biologics Inc. in China. While ChAdOX1 nCoV-19, Sputnik V, and Ad5-nCOV express the full-length spike protein, Ad26.COV2-S expresses a pre-fusion version [66]. Concerns have been raised about potential rare side effects, such as thrombotic thrombocytopenia and disseminated intravascular coagulation, associated with ChAdOx1 nCoV-19 and Ad26.COV2-S, particularly in young adults, despite the low incident rates. Ongoing investigations aim to address and mitigate these rare adverse reactions to ensure the safety and efficacy of these adenovirus vector vaccines [67,68,69]. Gene therapy, oncolytic virus therapy, immunotherapy, and development of vaccines are distinct approaches in the field of medical interventions, each with unique mechanisms and purposes.

### 3.1. Gene Therapy

Gene therapy typically involves the delivery of therapeutic genes using viral vectors or other delivery systems to correct genetic abnormalities or enhance cellular functions. It treats genetic disorders, corrects mutations, or enhances the body’s ability to fight disease. It targets genetic disorders at the root by introducing, altering, or replacing genes. It holds promise for a wide range of genetic diseases, including those with no existing cures. Challenges include efficient delivery methods and potential immune responses to the introduced genetic material. Currently, gene therapy modalities fall into five categories: gene-addition therapy, mRNA therapy, gene repair, mRNA repair, and cell therapy. These approaches are actively investigated, spanning pre-clinical testing to clinical trial phases [70]. Rather than employing DNA, mRNA molecules can be delivered to the airway cells in order to express the desired therapeutic protein [71]. Alternatively, mRNA repair approaches, also known as antisense therapies, can be employed. These involve administering short, single-stranded oligonucleotides to cells to target and repair the abnormal mRNA [72].

Medical research has produced viral and nonviral gene-delivery vectors. While viral vectors efficiently protect transgenes and cross cellular barriers, nonviral systems, such as cationic lipids and polymers, offer sustainable gene expression without triggering inflammatory or immune reactions, and are considered nontoxic [73]. The process of viral gene delivery involves integrating the transgene into viral DNA, which is then inserted into the viral vector. The vector binds to cell receptors, is internalized, and releases the capsid containing the modified DNA. The capsid moves to the nucleus, integrates the gene into the cell’s DNA, and enables protein production and gene expression [74]. Lentiviruses and most retroviruses follow this process [75]. Conversely, adenoviruses deliver genetic material to the cytoplasm or nucleus without integrating into the host genome [76]. Chemical nonviral vectors, including cationic polymers and lipids, are popular for gene delivery due to their safety, high transfection capacity, reduced toxicity, and easy preparation. These cationic molecules, like carbon nanotubes, quantum dots, and metal nanoparticles, form complexes with nucleic acids through electrostatic forces. They overcome barriers such as cellular uptake, endosomal escape, and gene delivery to the cytoplasm or nucleus, ensuring sustained gene expression [77].

Belonging to the Parvoviridae viral family, adeno-associated viruses (AAVs) are small, nonenveloped DNA viruses with a single strand. Although nonautonomous and unable to replicate independently, they integrate into the host genome in 0.1% of cases, primarily in chromosome 19. An intriguing characteristic of wild-type AAV is its ability to integrate its genome into a specific location on chromosome 19 (19q13-qter), known as the AAVS1 site [78,79]. Despite the lack of extensive homology between the AAVS1 site and the AAV genome, the presence of an active Rep binding element in AAVS1, along with the AAV Rep protein (a DNA helicase), enables site-specific integration of the vector. This Rep-mediated replication at AAVS1 may facilitate DNA repair processes that enable provirus integration [80]. Although only about 0.1% of wild-type AAV genomes integrate at AAVS1, this mechanism could still have implications in the virus’s natural history, as AAV proviruses have been found in AAVS1 following natural infection [78]. In some cases, integration has been observed in testicular tissue. Synthetic AAV vectors have also been engineered to integrate at the AAVS1 site through the Rep-dependent mechanism, but the inclusion of the rep gene in the vector can be detrimental due to its toxicity and impact on viral packaging capacity.

AAV vectors have low genotoxicity, making them safe for gene therapy, particularly in slow-dividing cells like cardiomyocytes, where their reduced transgene expression in dividing cells is a drawback [73]. AAVs also exhibit low immunogenicity and toxicity compared to other viral vectors, establishing them as safe and effective tools for gene therapy with minimal side effects and stable circulation in the bloodstream [81].

The approval of Glybera^®^ in Europe in 2012 marked a significant milestone for AAV gene therapy for in vivo viral vector treatment, but unfortunately, it was delisted after 4 years. By then only a single dose had been sold, making it impossible to keep its market registration. Despite the potential for small start-ups in clinical development, sustained progress necessitates partnerships and sufficient funding. Successful development hinges on strategic alliances, capital investments, and precise manufacturing and marketing strategies [54]. Despite the considerable therapeutic potential offered by gene therapy and gene-correction strategies, several challenges must be addressed before their widespread clinical adoption, particularly focusing on issues related to safety and efficacy [82].

### 3.2. Oncolytic Virus Therapy

Oncolytic virus therapy uses viruses that selectively infect and destroy cancer cells while sparing normal cells. Oncolytic viruses are engineered or naturally occurring viruses that replicate within cancer cells, causing their destruction. Oncolytic virus therapy targets and kills cancer cells directly, often in combination with other cancer therapies. It specifically targets and kills cancer cells, potentially reducing side effects compared to traditional treatments like chemotherapy. It shows promise in certain cancer types. Effectiveness can vary, and immune responses may limit repeated use.

Oncolytic Viruses (OVs) are a class of viruses that infect and replicate preferentially in cancer cells, leaving the normal cellular component that surrounds the tumor unharmed [83]. Oncolytic viruses (OVs) selectively infect and destroy cancer cells. Two main approaches exploit this selectivity: leveraging the natural affinity of viruses for tumors based on overexpressed markers or exploiting intracellular pathways and tumor-specific immune-avoidance mechanisms [84]. Viral infection occurs through interactions with cell surface receptors. OVs are chosen based on receptors overexpressed in cancer cells, such as CD155, a receptor upregulated in cancer cells that makes them an ideal target for oncolytic poliovirus [85]. In addition to the direct oncolytic effects mediated by viruses, OVs not only directly destroy cancer cells but also exert their anticancer effects by modulating the tumor microenvironment (TME) through various mechanisms.

OVs serve as a natural platform for recruiting CD8^+^ T cells and initiating innate and adaptive immune responses. Tumor lysis by OVs releases tumor-associated antigens (TAAs) and tumor neoantigens (TNAs), allowing dendritic cells (DCs) to present them, priming a specific anti-tumor T cell response. The release of viral epitopes induces antigenic presentation on MHC-I complexes, activating T cells directly [86]. OVs also induce immunogenic cell death (ICD), releasing danger-associated molecular patterns (DAMPs) like calreticulin (ecto-CRT), ATP, and HMGB1 [87]. Moreover, pathogen-associated molecular patterns (PAMPs) released during tumor cell burst activate the innate immune system through pattern-recognition receptors. Certain OVs, like vaccinia virus and HSV-1, exhibit antiangiogenic effects by infecting and destroying endothelial cells, enhancing overall efficacy [88].

Recent evidence indicates that viral infection can transform peripheral tissue into tertiary lymphoid structures (TLSs) resembling secondary lymphoid organs [89]. This conversion depends on type I interferon (IFN) production and the expression of CXCL13. In the context of cancer therapy, this may enhance the humoral response against tumor-specific antigens by recruiting B cells. Combining the oncolytic adenovirus (Ad5/3-E2F-d24-hTNFa-IRES-hIL2) TILT-123 with immune checkpoint inhibitors proved significantly more effective in treating ex vivo patient samples and demonstrated potent efficacy against in vivo tumor growth. Ovarian cancer, known for its low response rates to immune checkpoint inhibitors, could benefit from this approach. This study explores the impact of immune checkpoint inhibitor inhibition in the context of TILT-123 therapy for ovarian cancer [90].

T-VEC (talimogene laherparepvec), the sole FDA-approved oncolytic virus, is a modified herpes simplex 1 virus used for metastatic melanoma treatment [91]. It selectively replicates in melanoma cells, inducing lysis and producing GM-CSF to enhance systemic antitumor immune responses [91,92,93]. Clinical trials combining T-VEC with anti-PD-1 antibody, pembrolizumab, showed a 62% overall response rate with increased cytotoxic CD8^+^ T cells [94]. Adenoviruses (Ads) are versatile viral vectors, used as replicative oncolytic adenoviruses and replicative deficient adenoviruses for vaccines and gene therapy [31,95]. Ads exhibit genetic stability, a well-understood genome, and high in vivo transduction efficiency [74]. They can be engineered to bind different cell surface receptors. Ad’s nonenveloped nature and lytic replication cycle favor oncolysis [95]. Furthermore, Ad is a nonenveloped virus with a replication cycle culminating in lysis of host cells. This lytic replication cycle is favorable for oncolysis [96] when compared to enveloped viruses which complete replication by budding from intact, live, host cells. Ad-based therapy has demonstrated potent antitumor effects, infecting dendritic cells and upregulating immunostimulatory signals. Current studies explore Ads as cancer vaccines or treatments for infectious diseases like HIV, showcasing their versatility in cancer therapy [97]. 

Interleukin-12 (IL-12) is a cytokine commonly utilized to arm oncolytic viruses, such as adenoviruses (Ads). IL-12 promotes a robust antitumor effect by enhancing natural killer (NK) cell and cytotoxic T cell activities [98]. Ads armed with IL-12 have demonstrated improved antitumor effects and immune stimulation in both preclinical and clinical studies [99], including murine and Syrian hamster models [100]. The oncolytic adenovirus LOAd703 has shown efficacy in replicating and killing pancreatic cancer cells through oncolysis in vitro and in vivo. DNX-2440 (OAd-OX40L) is an armed oncolytic adenovirus designed for glioblastoma treatment, while Ad5/3-Δ24aCTLA4 and OAd-IFN-α are employed in advanced melanoma and pancreatic cancer therapy, respectively. Additionally, oncolytic adenoviruses with BiTE exhibit significantly enhanced antitumor activity in a lung cancer xenograft model [101] (Table 1).

### 3.3. Immunotherapy

The purpose of immunotherapy is to boost the body’s immune system to recognize and destroy cancer cells or pathogens. It includes various approaches like immune checkpoint inhibitors, adoptive cell therapy, and monoclonal antibodies, all designed to modulate or enhance immune responses. It is used in cancer treatment and for certain autoimmune diseases. It harnesses the body’s immune system to fight diseases, particularly in the context of cancer. It can lead to durable responses in some patients. Response rates can vary between individuals, and there may be immune-related side effects.

Cancer immunotherapy harnesses the patient’s immune system to fine-tune specific anti-tumor responses. Among various approaches, oncolytic viruses (OVs) have emerged as a novel form. OVs, whether natural or genetically modified, selectively kill cancer cells and induce immunogenic cancer cell death. This process releases tumor antigens, triggering anti-tumor immunity, essentially turning OVs into in situ therapeutic cancer vaccines. OVs can also be engineered for intratumoral delivery of immunostimulatory molecules, enhancing the anti-tumor response. Furthermore, OVs can synergize with other cancer immunotherapies, including immune checkpoint inhibitors and CAR-T cells [83].

Nowadays, interest in using regulatory T cells (Tregs) for adoptive cell therapy in autoimmune diseases and transplant rejection has grown. Despite advancements, Treg therapies indiscriminately suppress the immune system without offering permanent disease stabilization. Quantitative and qualitative changes in Tregs and effector T cells have been observed in over eighty autoimmune disorders. The past decade has seen the development of small-molecule inhibitors and humanized monoclonal antibodies, providing relief for autoimmune disease symptoms. However, these treatments often lead to drug resistance and toxicities due to the need for lifelong administration. Genetically engineered Tregs in adoptive cell therapy present a promising alternative, offering greater specificity, longevity, and tissue regeneration [102].

In recent years, viral vector gene therapy has experienced significant progress, particularly with the use of adenoviruses. Adenovirus has become a popular option for cancer gene therapy. These vectors can be divided into two categories: replication-deficient viruses and replication-competent oncolytic viruses. The former acts as a delivery system for vaccines and gene therapy, while the latter has been specifically designed to target and eliminate cancer cells. By releasing tumor antigens and inducing local inflammation, oncolytic adenoviruses effectively convert ‘cold’ tumors to ‘hot’ tumors, strengthening the body’s defenses against cancer. The combination of oncolytic viruses and immunotherapy shows great potential in cancer treatment [101].

One study investigates an adenoviral vector, Ad5 [E1-, E2b-], carrying the carcinoembryonic antigen (CEA) gene to induce cell-mediated immunity (CMI) against tumors. In Ad5 pre-immunized mice, repeated immunizations with Ad5 [E1-, E2b-]-CEA were compared to current generation Ad5 [E1-]-CEA for CEA immunogenicity and in vivo anti-tumor effects. Ad5 [E1-, E2b-]-CEA induced significantly increased CEA-specific CMI responses in Ad5-immune mice compared to Ad5 [E1-]-CEA. Moreover, Ad5 [E1-, E2b-]-CEA treatment in Ad5-immune mice with CEA-expressing tumors led to an enhanced anti-tumor response compared to Ad5 [E1-]-CEA-treated mice [103] (Table 1). 

### 3.4. Vaccine Development

Vaccines work by activating the immune system to identify and remember harmful pathogens, shielding against potential infections. By introducing harmless elements of a pathogen, such as proteins or weakened viruses, the immune system is triggered to respond without causing illness. This proactive approach primes the immune system to identify and battle specific pathogens, effectively preventing the spread of infectious diseases. The use of vaccines has played a vital role in managing and eradicating various diseases. However, not all illnesses have vaccine options, and the effectiveness may vary depending on factors like a pathogen’s ability to mutate. Each year, vaccines save the lives of over 15 million people. Through a highly effective worldwide vaccination effort, smallpox was completely eliminated in 1980, followed by rinderpest in 2011 [104]. 

Creating durable and effective vaccines to combat a variety of viral pathogens presents considerable obstacles. These challenges arise from heightened mutation rates and antigenic diversity, as well as constraints on the immunogenicity of conserved epitopes and the technical complexities of producing structurally accurate immunogens [26].

To develop an adenovirus vector-based vaccine, various modifications to the fiber protein have been explored. One approach involves constructing an adenovirus library by displaying random peptides on the fiber knob to generate vectors targeting specific cell types [105]. Additionally, chimeric adenovirus vectors with modified fiber knobs have been developed to evade pre-existing immunity and enhance immunogenicity [106,107,108]. Furthermore, replication-deficient chimpanzee adenovirus-vectored vaccines expressing conserved antigens have been constructed, demonstrating the potential for utilizing different adenovirus species for vaccine development [109,110]. Scientists tested a vaccine on rhesus monkeys using Ad5/35 and modified vaccinia virus Ankara (MVA) vectors expressing simian immunodeficiency virus (SIV) antigens (gag and gp120). They also determined the effectiveness of this vaccine regimen in protecting against SIVmac239 infection in rhesus monkeys. The study found that this vaccine regimen significantly decreased SIV RNA and DNA viral loads. Vaccinated monkeys maintained central memory CD4 T-cell numbers, suggesting reduced susceptibility to retrovirus infection. SIV and chimeric SHIV were used to model HIV in monkeys [111].

Moreover, the tropism of adenovirus vectors has been broadened through fiber knob serotype switching, allowing for enhanced targeting of specific cell types [112]. However, the lack of cell-type specific ligands for targeted adenovirus vectors has been identified as a limitation, hindering the wider application of fiber-modified adenovirus vectors for targeted therapies [113]. Additionally, modifications to the antigen used in adenovirus vaccines have been shown to improve the induced T-cell response, highlighting the potential for further advancements in vaccine development [114]. The development of adenovirus vector-based vaccines involves various modifications to the fiber protein, including the display of random peptides, chimeric vector construction, and tropism broadening. These modifications aim to enhance vector targeting, immunogenicity, and evasion of pre-existing immunity, ultimately contributing to the advancement of adenovirus vector-based vaccine platforms.

Adenoviral vector vaccines are presently undergoing human clinical trials for cancer, with a focus on carcinoembryonic antigen (CEA), a glycoprotein found in various cancers. One such vaccine, ETBX-011, is an engineered adenoviral vaccine that carries a modified CEA featuring the highly immunogenic epitope CAP1-6D. This vaccine has demonstrated the ability to trigger CEA-specific cell-mediated immune responses, showcasing promising anti-tumor activity [115]. ETBX-011 is designed with deletions in the E1 genes (E1A and E1B), the E2B gene (encoding the viral polymerase), and the E3 genes [103,116]. The expression of carcinoembryonic antigen (CEA) is controlled by the cytomegalovirus (CMV) promoter. This configuration leads to robust CEA expression, eliciting specific cell-mediated immune responses against CEA and demonstrating notable anti-tumor activity.

Ad5-PSA is a non-replicating adenovirus type 5 that contains the human prostate-specific antigen (PSA) gene. In a mouse model of prostate cancer, this adenovirus vaccine prompted robust immune responses against PSA and led to the elimination of tumor cells producing PSA [117].

In a phase I clinical trial for HPV-positive cancers, a replication-deficient adenovirus vaccine, Ad-E6E7 [118], is combined with the oncolytic maraba virus strain MG1 vaccine, MG1-E6E7, along with sequential atezolizumab treatment (anti-PD-L1 antibody). Both Ad-E6E7 and MG1-E6E7 express modified human papillomavirus (HPV) genes E6 and E7, removing regions necessary for oncogenic transformation [119]. In preclinical studies, Ad-E6E7 priming followed by MG1-E6E7 boosting resulted in remarkably potent tumor-specific immune responses, significantly extending survival in various mouse cancer models [120]. 

Furthermore, two clinical trials investigate viral vector-based cancer vaccines for advanced MAGE-A3-positive solid tumors. The first trial involves oncolytic maraba MG1-MAGEA3 alone or after adenovirus-MAGEA3 treatment. The second trial evaluates oncolytic MG1-MAGEA3 with Ad-MAGEA3 in combination with pembrolizumab (anti-PD-1 antibody) for previously treated metastatic non-small cell lung cancer. In summary, viral vector-based cancer vaccines, including adenovirus-based ones, offer promising therapeutic approaches to trigger antitumor immune responses [101,121,122].

It is challenging to determine a ‘best’ approach among gene therapy, oncolytic virus therapy, immunotherapy, and vaccines, as each serves different purposes and is tailored to specific medical needs. The effectiveness of these approaches depends on the nature of the condition being treated and the individual patient’s characteristics [123]. In summary, while gene therapy focuses on modifying genetic material, oncolytic virus therapy targets cancer cells specifically, immunotherapy harnesses the immune system to combat diseases, and vaccines prime the immune system to prevent future infections. These approaches often complement each other in the broader landscape of modern medicine (Table 1).

**Table 1 viruses-16-01094-t001:** Different types of Adv vector with their specialty, advantages along with clinical usages.

Sl. No.	Generation/Type	Name of the Vector	Specialty	Advantages	Clinical Use	References
1.	Wild-Type Adenovirus (WTAd)	Adv2, Adv5, Adv11, Adv26	Natural, unmodified adenoviruses Selective cancer-killing agents Understanding regulation of gene expression by microRNA	High cloning capacity, short expression time, and comparatively high immune response	Vaccination, oncolytic therapy, virotherapy, and gene therapy	[124,125]
2.	First-Generation Adenovirus Vectors (FGAd)	Modified WTAd with essential genes deleted. e.g., Ad5ΔE1 or Ad5ΔE1, E3	Deleting the E1 region and the E3 region from the adenoviral genome. Human Embryonic Kidney (HEK) 293 cell is commonly used	High titer level; very efficient transduction of most cells and tissues	Vaccination, and anti-cancer therapy	[74,126]
3.	Second-Generation Adenovirus Vectors (SGAd)	Modified fiber and hexon	Deletion of the E2 and E4 regions from the adenoviral genome	Deletions significantly reduce the synthesis of adenoviral proteins and SGAd still induces host immune responses	Vaccination	[126]
4.	Helper-Dependent Adenovirus Vectors (HDAd)	HDAd5 with extended deletions HDAd5/35++ HDAd6/35++	Large-scale deletion of viral genes Long-term transgene expression without chronic toxicity	Vectors are less toxic and have much larger DNA carrying capacity Safe and good gene expression Efficient	Lung cancer gene therapy Cystic fibrosis Genetic disorder Liver directed gene therapy In vivo gene transfer	[127,128,129,130,131]

## 4. Conclusions

Adenoviral vectors have been studied extensively in several clinical trials since their first use as gene-delivery vehicles. Challenges such as toxicity, pre-existing immunity in hosts, and complexity related to construction are continually being addressed. Optimization of adenovirus vectors is vital for both experimental research and therapeutic applications. Regulatory compliance, scalability, long-term stability, and versatility are also achieved through optimization, facilitating the development of effective and safe gene therapies. Furthermore, a bulk of information regarding the biology of adenoviral vectors and the immune responses they elicit in animals has been extremely helpful in the development of potent vaccine candidates against several viral infections that have progressed to advanced clinical stages. Adenoviral vectors have been approved for use in cancer and gene therapy in humans, in addition to infectious diseases. As a result, adenoviral vectors have created new opportunities for oncolytic virus therapy, vaccine antigen delivery, and gene therapy.

## Figures and Tables

**Figure 1 viruses-16-01094-f001:**
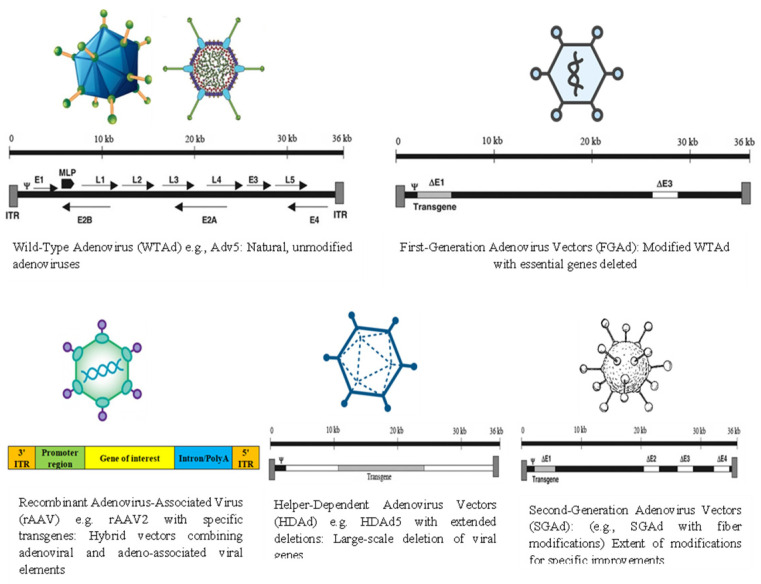
Wild-Type adenovirus and different genetically modified adenovirus vectors.

**Figure 2 viruses-16-01094-f002:**
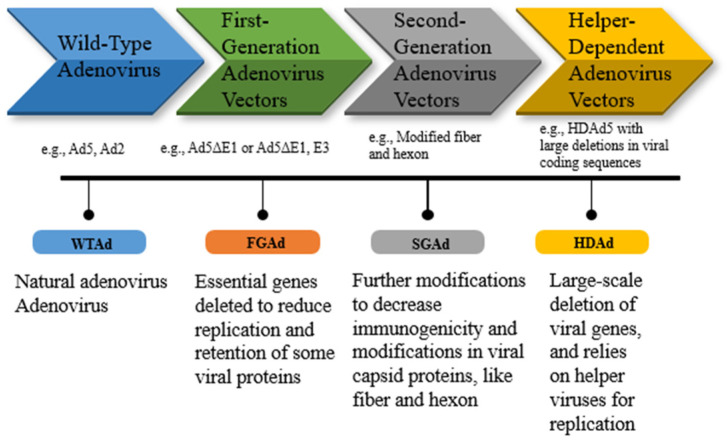
Emergence of different adenovirus vectors with their genetic modifications.

## Data Availability

Not applicable.

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
