# Peer review of "Clinical Application of Adenovirus (AdV): A Comprehensive Review"

_viruses, 2024, doi:10.3390/v16071094_

Round 1

Reviewer 1 Report

Comments and Suggestions for Authors

The manuscript by Salauddin and collaborators reviews adenoviruses, their biology, the vectors and the clinical applications based on adenovirus-mediated gene transfer. The manuscript also reviews the some of the approaches taken in the clinical applications involving the administration with a therapeutic intent of adenoviruses to patients.

The manuscript is well written but needs some careful check of the statements made.

Below a number of examples that need attention.

1)      Line 21: it is written: ‘Adenovirus vectors have the potential to revolutionize gene therapy by modifying wild-type viruses to render them replication-defective.’ This is not at all unique to adenoviruses as many of the virus families can be made replication defective. It is therefor not justified state that adenoviruses can ‘revolutionize gene therapy’. Please tune down this statement.

2)      Line 33: It is not a unique statement of adenoviruses to have a genetic makeup of double-stranded DNA. Also Hepadnaviridae, Polyomaviridae, Papillomaviridae, Adenoviridae, Herpesviridae, Poxviridae and Asfarviridae have double-stranded DNA genomes. Please correct.

3)      Line 34: The family of adenoviruses contains many more than 7 distinct species.  The authors probably refer to human adenoviruses. Please correct.

4)      Line 38: The testing of human samples yielded at least 116 ‘types’ of human adenoviruses, which consist of a much larger number of number of isolates (see: human adenovirus working group, http://hadvwg.gmu.edu/ data of March 2024)

5)      Line 42:  The statement; ’Adenoviral vectors have distinct features such as they can deliver many copies of the genome into one cell’ is confusing, as this is possible for most, if not all viral vector systems. Please amend to clarify.

6)      Line 46: why do the propensities to provoke strong immune responses and to yield transient gene expression make these vectors useful for treating single gene disorders? This rather seems a counter indication. Please clarify.

7)      Line 75: The citation of ref. 12 seems to be erroneous as this paper refers to AAV.

8)      The text at line 75: ‘Enhanced transduction efficiency ensures effective delivery of genetic material to target cells, while minimizing immunogenicity is crucial for avoiding immune responses. Optimization enables targeted tropism, directing vectors to specific cell types, and addresses safety concerns associated with toxicity [13].’ Seems to refer to data obtained with AAV, not adenovirus. Please correct.   

9)      Line 81: A transposon-based system, employing Sleeping Beauty, optimized HEK293 cells for

chimpanzee Adenovirus (chAd) vector production, a Human Mastadenovirus C species.  Please explain this statement, add a reference, and mention which chimpanzee adenovirus is referred to. This is important as there are also chimpanzee adenoviruses that are not HAdV-C.

10)   Line 83: Please clarify what is intended by line 83: Swapping hAd5 E1 with chAd-C E1 via CRISPR/Cas9 revealed chAd-C E1 alone did not support HEK293 survival. As adenovirus replication in 293 is lytic this is what is desirable isn’t it?! Please clarify what is intended here.

11)   Paragraph 2.1 The term RCA is conventionally used for replication-competent adenoviruses that are generated during production by the vector acquiring E1 functions from 293 cells. For adenoviruses that replicate preferentially in tumor cells the term CRAds is used (Conditionally Replicating Adenoviruses). I would suggest the authors do the same here.

12)   Line 187: What distinguishes a ‘modification replication-deficient Ad’ from a classical ‘replication deficient Ad’ or a ‘first generation Adenovirus Vector’? Please clarify.

13)   Line 410: What is mRNA repair and what is mRNA therapy? Please define these terms.

14)   Line 412: the statements: ‘While viral vectors efficiently protect transgenes and cross cellular barriers, nonviral systems, such as cationic lipids and polymers, offer sustainable gene expression without triggering inflammatory or immune reactions, and are considered nontoxic [64]’ is very intriguing. Could the authors go into some details? Which cellular barriers are passed by adenoviruses? Which cationic lipids and polymers offer sustainable gene expression. Some detail more would be appreciated.

15)   Line 417: The statement that: ’AAV integrates in 0.1% of the cases integrate into the host cell genome, preferentially in chromosome 19.’ needs more detail. How is the integration efficiency of 0.1% defined. Is this 0.1% of the input genomes? The preferential integration into chromosome 19 may not be correct. Integration in this locus is dependent on the AAV rep protein. This protein should not be expressed in state of the art AAV vector preps, as it comes from wtAAV genomes.

16)   Line 424: It would be prudent to state that although Glybera received marked approval in 2012, it was delisted after 4 years. By then only a single dose had been sold, making it impossible to keep its marked registration.   

17)   Table 1: Why is AAV covered in rows 5 and 6? The AAV virus has really nothing to do with adenovirus?! I would remove these data.

18)   Would it be possible to include a list of adenoviruses that received marked approval in this review? That would be informative for readers.

19)   Reference 23 is incomplete.

Comments on the Quality of English Language

The language is suffienct, but would bennefit from careful proofreading. 

Author Response

Response to reviewer 1

The manuscript is well written but needs some careful check of the statements made. Below a number of examples that need attention.

Response: Thank you very much for your appreciation, valuable comments, which significantly improved the quality of our manuscript.

  • Line 21: it is written: ‘Adenovirus vectors have the potential to revolutionize gene therapy by modifying wild-type viruses to render them replication-defective.’ This is not at all unique to adenoviruses as many of the virus families can be made replication defective. It is therefore not justified state that adenoviruses can ‘revolutionize gene therapy’. Please tune down this statement.

Answer: Thanks for the suggestion and now we fine-tune the statement and delete the word unique from the statement (Line: 21, Page: 01).

  • Line 33: It is not a unique statement of adenoviruses to have a genetic makeup of double-stranded DNA. Also Hepadnaviridae, Polyomaviridae, Papillomaviridae, Adenoviridae, Herpesviridae, Poxviridae and Asfarviridae have double-stranded DNA genomes. Please correct.

Answer: We have corrected the statement as “Adenoviruses belong to a prominent family of adenoviridae, non-enveloped viruses with a genetic makeup of double-stranded DNA that commonly result in respiratory illnesses in individuals of all ages”.(Line: 33-34, page: 01).

  • Line 34: The family of adenoviruses contains many more than 7 distinct species.  The authors probably refer to human adenoviruses. Please correct.

Answer: Thanks for the nice suggestion. We have now corrected the sentence as “Human adenoviruses consists of seven distinct species (A to G) and can also elicit a range of other health complications including gastrointestinal, ocular, urinary, and neurological disorders”. (Line: 34, page: 01).

  • Line 38: The testing of human samples yielded at least 116 ‘types’ of human adenoviruses, which consist of a much larger number of number of isolates (see: human adenovirus working group, http://hadvwg.gmu.edu/ data of March 2024).

Answer: Thanks for advising us on the updated types of human adenoviruses. Now we have updated the information as “The testing of human samples yielded at least 116 ‘types’ of human adenoviruses, which consist of a much larger number of isolates” (Line: 38-41, page: 01).

5)      Line 42:  The statement; ’Adenoviral vectors have distinct features such as they can deliver many copies of the genome into one cell’ is confusing, as this is possible for most, if not all viral vector systems. Please amend to clarify.

Answer: Now we elaborated and explained our statement in the article (Line: 44-52, page: 01-02) as follows-

Over the past three decades, viral vectors such as AdVs, adeno-associated viruses (AAVs), herpes simplex virus (HSV), lentivirus, and retrovirus have been extensively investigated for their applications in gene therapy and vaccine development. Retroviruses integrate genetic material into infected cells, but they require ongoing cell division, which is a significant limitation when used as a viral vector. Lentiviral vectors, such as human immunodeficiency virus (HIV), have a higher chance of unpredictable genome integration in the host genome. Adeno-associated virus vectors also infect both dividing and non-dividing cells with a limited immune response, but they have size constraints in terms of cargo capacity (maximum 4.7 kb). In contrast, adenovirus vectors have several distinct features, including: (1) the ability to deliver multiple copies of the transgenes into a single host cell, resulting in high gene expression; (2) transient gene expression because the DNA cargo remains episomal and does not integrate into the host genome; (3) the capability to transduce both dividing and non-dividing cells; and (4) strong immunogenicity compared to many other vectors.

6)      Line 46: why do the propensities to provoke strong immune responses and to yield transient gene expression make these vectors useful for treating single gene disorders? This rather seems a counter indication. Please clarify.

      Answer: The use of HC-Ad was able to overcome some of the limitations of first and second-generation vectors and was employed in some strategies. In nonhuman primate models, the expression of the baboon alpha-fetoprotein transgene delivered by a HC-Ad persisted up to 7 years without adverse effects, declining to about 10%/year (Brunetti-Pierri et al., 2013). In a mouse model of primary kidney disease hyperoxaluria type 1, HC-Ad transferred the alanine-glyoxylate aminotransferase gene under control of a liver-specific promoter, improving the clinical condition of the animals for at least 24 weeks (Castello et al., 2016). In another study, primary dystrophin-deficient mouse myoblasts were successfully transduced with an adenoviral vector carrying the full-length murine dystrophin cDNA under control of a muscle-specific promoter and a lacZ reporter construct (Kochanek et al., 1996).

7)      Line 75: The citation of ref. 12 seems to be erroneous as this paper refers to AAV.

Answer: Now we have changed the reference.

8)      The text at line 75: ‘Enhanced transduction efficiency ensures effective delivery of genetic material to target cells, while minimizing immunogenicity is crucial for avoiding immune responses. Optimization enables targeted tropism, directing vectors to specific cell types, and addresses safety concerns associated with toxicity [13].’ Seems to refer to data obtained with AAV, not adenovirus. Please correct.

Answer: The statements about enhanced transduction efficiency, minimizing immunogenicity, enabling targeted tropism, and addressing safety concerns associated with toxicity are true for both Adeno-associated virus (AAV) and adenovirus vectors. 

The optimization of both adeno-associated virus (AAV) and adenovirus vectors involves enhancing transduction efficiency, minimizing immunogenicity, enabling targeted tropism, and addressing safety concerns associated with toxicity. Enhanced transduction efficiency ensures effective delivery of genetic material to target cells, while minimizing immunogenicity is crucial for avoiding immune responses, particularly in the case of adenovirus vectors which are inherently more immunogenic than AAV. Both types of vectors can be engineered to target specific cell types, improving targeted delivery and reducing off-target effects. Safety concerns, such as toxicity and potential immune responses, are addressed through various modifications, including making the vectors more replication-deficient and reducing the expression of viral proteins. These optimizations are vital for the effectiveness and safety of gene therapy and other therapeutic applications (Wold & Toth, 2013; Li & Samulski, 2020).

References

Wold, W. S. M., & Toth, K. (2013). Adenovirus vectors for gene therapy, vaccination and cancer gene therapy. Current Gene Therapy, 13(6), 421-433. doi:10.2174/1566523213666131129125607

Li, C., & Samulski, R. J. (2020). Engineering adeno-associated virus vectors for gene therapy. Nature Reviews Genetics, 21(4), 255-272. doi:10.1038/s41576-019-0205-0

9)      Line 81: A transposon-based system, employing Sleeping Beauty, optimized HEK293 cells for

chimpanzee Adenovirus (chAd) vector production, a Human Mastadenovirus C species.  Please explain this statement, add a reference, and mention which chimpanzee adenovirus is referred to. This is important as there are also chimpanzee adenoviruses that are not HAdV-C.

Answer:  We have corrected the statement and added a reference also (Line: 97-101, page: 02-03).

10)   Line 83: Please clarify what is intended by line 83: Swapping hAd5 E1 with chAd-C E1 via CRISPR/Cas9 revealed chAd-C E1 alone did not support HEK293 survival. As adenovirus replication in 293 is lytic this is what is desirable isn’t it?! Please clarify what is intended here.

Answer: In the absence of human Ad5 E1, chimp Ad-C E1 gene did not support HEK293 survival. To improve chAd-C vector production, HEK293 cells was engineered for stable expression of the chAd-C precursor terminal protein (ch.pTP), which plays a crucial role in chimpanzee Adenoviral DNA replication. The introduction of exogenous ch.pTP expression has a substantial positive impact on the packaging and amplification of recombinant chAd-C vectors. Therefore, the modified HEK293ch.pTP cells may serve as a more effective cell line for producing these vectors.

11)   Paragraph 2.1 The term RCA is conventionally used for replication-competent adenoviruses that are generated during production by the vector acquiring E1 functions from 293 cells. For adenoviruses that replicate preferentially in tumor cells the term CRAds is used (Conditionally Replicating Adenoviruses). I would suggest the authors do the same here.

Answer: We have added a new paragraph regarding the CRAds (Line: 156-165, page: 04).

12)   Line 187: What distinguishes a ‘modification replication-deficient Ad’ from a classical ‘replication deficient Ad’ or a ‘first generation Adenovirus Vector’? Please clarify.

Answer: Thanks for the concern. Our explanation is as follows

Classical Replication-Deficient Adenovirus (First Generation Adenovirus Vector)

Classical replication-deficient adenovirus vectors, or first-generation adenovirus vectors, are characterized by deletions in the E1 and E3 regions of the adenovirus genome. The E1 deletion makes the virus unable to replicate in normal human cells by removing essential genes required for viral replication. The E3 deletion, which is not necessary for replication in cell culture, provides additional space for the insertion of therapeutic genes. These vectors are produced in cell lines such as HEK293 that supply the missing E1 proteins, allowing the virus to replicate during production but not in the patient. Despite their utility, these vectors can still provoke an immune response due to low-level expression of viral proteins and pose a risk of recombination with wild-type adenovirus.

Modification Replication-Deficient Adenovirus

Modification replication-deficient adenovirus vectors involve more advanced genetic modifications beyond the E1 and E3 deletions. These vectors may include additional deletions or mutations in other essential genes, such as E2 and E4, to further ensure replication deficiency and enhance safety. They are engineered to minimize the expression of viral proteins that could trigger immune responses, making them less immunogenic and suitable for repeated administration. These modifications also reduce the risk of recombination events and allow for greater flexibility in delivering larger or more complex therapeutic genes, optimizing the vector's capacity and therapeutic potential.

13)   Line 410: What is mRNA repair and what is mRNA therapy? Please define these terms.

Line: 441-446

Answer: Rather than employing DNA, mRNA molecules can be delivered to the airway cells in order to express the desired therapeutic protein [Sahu et al., 2019]. Alternatively, mRNA repair approaches, also known as antisense therapies, can be employed. These involve administering short, single-stranded oligonucleotides to cells to target and repair the abnormal mRNA [Martinovich et al., 2018].

14)   Line 412: the statements: ‘While viral vectors efficiently protect transgenes and cross cellular barriers, nonviral systems, such as cationic lipids and polymers, offer sustainable gene expression without triggering inflammatory or immune reactions, and are considered nontoxic [64]’ is very intriguing. Could the authors go into some details? Which cellular barriers are passed by adenoviruses? Which cationic lipids and polymers offer sustainable gene expression. Some detail more would be appreciated.

Answer: Line: 449-462

The process of viral gene delivery involves integrating the transgene into viral DNA, which is then inserted into the viral vector. The vector binds to cell receptors, is internalized, and releases the capsid containing the modified DNA. The capsid moves to the nucleus, integrates the gene into the cell's DNA, and enables protein production and gene expression [Lee et al., 2017]. Lentiviruses and most Retroviruses follow this process [Desfarges and Ciuffi, 2012]. Conversely, Adenoviruses deliver genetic material to the cytoplasm or nucleus without integrating into the host genome [Bulcha et al., 2021]. Chemical nonviral vectors, including cationic polymers and lipids, are popular for gene delivery due to their safety, high transfection capacity, reduced toxicity, and easy preparation. These cationic molecules, like carbon nanotubes, quantum dots, and metal nanoparticles, form complexes with nucleic acids through electrostatic forces. They overcome barriers such as cellular uptake, endosomal escape, and gene delivery to the cytoplasm or nucleus, ensuring sustained gene expression [Wang et al., 2013].

15)   Line 417: The statement that: ’AAV integrates in 0.1% of the cases integrate into the host cell genome, preferentially in chromosome 19.’ needs more detail. How is the integration efficiency of 0.1% defined. Is this 0.1% of the input genomes? The preferential integration into chromosome 19 may not be correct. Integration in this locus is dependent on the AAV rep protein. This protein should not be expressed in state of the art AAV vector preps, as it comes from wtAAV genomes.

Answer: Line: 465-477

An intriguing characteristic of wild-type AAV is its ability to integrate its genome into a specific location on chromosome 19 (19q13-qter), known as the AAVS1 site [McCarty et al., 2001; Samulski et al., 1991]. Despite the lack of extensive homology between the AAVS1 site and the AAV genome, the presence of an active Rep binding element in AAVS1, along with the AAV Rep protein (a DNA helicase), enables site-specific integration of the vector. This Rep-mediated replication at AAVS1 may facilitate DNA repair processes that enable provirus integration [Kotin et al., 1992]. Although only about 0.1% of wild-type AAV genomes integrate at AAVS1, this mechanism could still have implications in the virus's natural history, as AAV proviruses have been found in AAVS1 following natural infection [McCarty et al., 2004]. In some cases, integration has been observed in testicular tissue. Synthetic AAV vectors have also been engineered to integrate at the AAVS1 site through the Rep-dependent mechanism, but the inclusion of the rep gene in the vector can be detrimental due to its toxicity and impact on viral packaging capacity.

16)   Line 424: It would be prudent to state that although Glybera received marked approval in 2012, it was delisted after 4 years. By then only a single dose had been sold, making it impossible to keep its marked registration. 

Answer: We have added the suggested statement (Line: 483-487)  

17)   Table 1: Why is AAV covered in rows 5 and 6? The AAV virus has really nothing to do with adenovirus?! I would remove these data.

Answer: We have deleted the rows 5 and 6 in Table 1.

18)   Would it be possible to include a list of adenoviruses that received marked approval in this review? That would be informative for readers.

Answer: Thank you for your valuable suggestion. We believe that adding a list as you suggested would indeed enrich our article. However, a recently published book chapter has already discussed this topic in detail. Therefore, we have decided not to review it again in our article.

Reference:

Singh, S., Kumar, R., & Agrawal, B. (2019). Adenoviral vector-based vaccines and gene therapies: current status and future prospects. Adenoviruses4, 53-91.

19)   Reference 23 is incomplete.

Answer: Corrected

Reviewer 2 Report

Comments and Suggestions for Authors

1.       Missing references, lines 241-263; lines 608-613. It will be suitable to provide original research articles, not only to use the review papers, for the three generations of Ads and other important studies.

2.       In section 2.1. Replication Competent Adenovirus, the authors indicated in line 101-102: “In gene therapy, replication-competent adenoviruses can be engineered to deliver therapeutic genes to target cells, taking advantage of the virus's natural ability to efficiently enter cells.”  It is not clear what is the advantage of such vectors, as all other Ad vectors can also effectively enter cells and “the use of replication-competent adenoviruses poses safety concerns”.

3.       Sections 2.2 is Replication deficient Ad and 2.3 is Modification Replication Deficient Ad. It is not clear what is the difference between the two groups of Ads. The authors stated that in line 175-176: “a "modification replication-deficient adenovirus" refers to an adenovirus that has been genetically altered to disable its ability to replicate efficiently within host cells”; in line 183-184: “A "replication-deficient adenovirus" refers to an adenovirus that has undergone genetic modifications to render itself incapable of replicating efficiently within host cells”. One is “disable” and another is “incapable”. Is there any difference between them.

4.       In table 1, first and second generations of Ads are not for oncolytic therapies.

5.       Line 615-623 is a well summary, but this does not belong to section 3.4. Vaccine Development.

6.       Minors that need check. Line 63-64: “which poses a major obstacle for the effectiveness of HAdV vectors [10] Similarly, administration of an AdV vector provokes…”

Line 223-224: “They modified the Ad5 by adding a key component (ELDKWA) 223 known for broadly neutralizing various HIV-1 strains. Additionally, we included…”

Comments on the Quality of English Language

Need some editings. 

Author Response

Response to Reviewer- 2

Thank you very much for your appreciation, valuable comments, which significantly improved the quality of our manuscript.

  1. Missing references, lines 241-263; lines 608-613. It will be suitable to provide original research articles, not only to use the review papers, for the three generations of Ads and other important studies.

Line: 279, page: 06, ref. 43, 44

Mizuguchi, H., & Hayakawa, T. (2002). Adenovirus vectors containing chimeric type 5 and type 35 fiber proteins exhibit altered and expanded tropism and increase the size limit of foreign genes. Gene285(1-2), 69-77.

Yang, M., Yang, C. S., Guo, W., Tang, J., Huang, Q., Feng, S., ... & Liu, Y. Q. (2017). A novel fiber chimeric conditionally replicative adenovirus-Ad5/F35 for tumor therapy. Cancer Biology & Therapy18(11), 833-840.

after line 274 –

Adenoviral vectors present certain characteristics that impact their application. Firstly, the vector's DNA does not integrate into the host cell genome but persists as episomal DNA, particularly prone to loss in dividing cells (Thomas et al., 2003). Additionally, the live virus generated by adenoviral vectors triggers a robust immune response in animals, limiting its in vivo utility (Cheng et al., 2007). Furthermore, utilizing adenoviral vectors involves intricate processes, including live virus production in packaging cells and the measurement of viral titer, making it technically demanding and time-consuming (Parker et al., 2006). Key elements in the vector include the 5' inverted terminal repeat (ITR), 3' ITR, Ψ packaging signal, promoters, Kozak consensus sequence, open reading frame (ORF), polyadenylation signals, and markers, each serving specific functions in facilitating gene expression and vector replication (Mittereder et al., 1996). The vector design also incorporates the ΔAd5/F35 configuration, enhancing tropism for cells with low or lacking coxsackievirus and adenovirus receptor expression (Shayakhmetov et al., 2000). The plasmid backbone, containing an ampicillin resistance gene and a PacI restriction site, ensures efficient vector linearization for packaging (Morsy et al., 1998).

line 282, ref. 45 –

Thomas, C. E., Ehrhardt, A., & Kay, M. A. (2003). Progress and problems with the use of viral vectors for gene therapy. Nature Reviews Genetics, 4(5), 346-358. https://doi.org/10.1038/nrg1066

line 284, ref. 46 –

Cheng, C., Gall, J. G., Kong, W. P., Sheets, R. L., Gomez, P. L., King, C. R., & Nabel, G. J. (2007). Mechanism of Ad5 vaccine immunity and toxicity: Fiber shaft targeting of dendritic cells. PLoS Pathogens, 3(2), e25. https://doi.org/10.1371/journal.ppat.0030025

line 286, ref. 47 –

Parker, A. L., Waddington, S. N., Nicol, C. G., Shayakhmetov, D. M., Buckley, S. M., Denby, L., ... & McVey, J. H. (2006). Multiple vitamin K-dependent coagulation zymogens promote adenovirus-mediated gene delivery in vitro and in vivo. Journal of Virology, 80(14), 6824-6830. https://doi.org/10.1128/JVI.00103-06

line 290,ref. 48 –

Mittereder, N., March, K. L., & Trapnell, B. C. (1996). Evaluation of the concentration and bioactivity of adenovirus vectors for gene therapy. Journal of Virology, 70(11), 7498-7509. https://jvi.asm.org/content/70/11/7498

line 291, ref. 49 –

Shayakhmetov, D. M., Papayannopoulou, T., Stamatoyannopoulos, G., & Lieber, A. (2000). Efficient gene transfer into human CD34+ cells by a retargeted adenovirus vector. Journal of Virology, 74(6), 2567-2583. https://doi.org/10.1128/JVI.74.6.2567-2583.2000

line 293, ref. 50 –

Morsy, M. A., Gu, M., Motzel, S., Zhao, J., Lin, J., Su, Q., ... & Gao, G. P. (1998). An adenoviral vector deleted for all viral genes results in highly efficient gene delivery: Application in cancer gene therapy. Human Gene Therapy, 9(15), 2259-2273. https://doi.org/10.1089/hum.1998.9.15-2259

In line 675, ref. 101, 121, 122.

Sato-Dahlman, M., LaRocca, C. J., Yanagiba, C., & Yamamoto, M. (2020). Adenovirus and immunotherapy: advancing cancer treatment by combination. Cancers12(5), 1295.

Jonker, D. J., Hotte, S. J., Abdul Razak, A. R., Renouf, D. J., Lichty, B., Bell, J. C., ... & Dancey, J. (2017). Phase I study of oncolytic virus (OV) MG1 maraba/MAGE-A3 (MG1MA3), with and without transgenic MAGE-A3 adenovirus vaccine (AdMA3) in incurable advanced/metastatic MAGE-A3-expressing solid tumours: CCTG IND. 214.

Pol, J. G., Acuna, S. A., Yadollahi, B., Tang, N., Stephenson, K. B., Atherton, M. J., ... & McCart, J. A. (2019). Preclinical evaluation of a MAGE-A3 vaccination utilizing the oncolytic Maraba virus currently in first-in-human trials. Oncoimmunology8(1), e1512329.

  1. In section 2.1. Replication Competent Adenovirus, the authors indicated in line 101-102: “In gene therapy, replication-competent adenoviruses can be engineered to deliver therapeutic genes to target cells, taking advantage of the virus's natural ability to efficiently enter cells.”  It is not clear what is the advantage of such vectors, as all other Ad vectors can also effectively enter cells and “the use of replication-competent adenoviruses poses safety concerns”.

Line: 116-123, page: 03

Answer: We rewrite our statement as follows-

Replication-competent (oncolytic) adenovirus vectors are employed for cancer gene therapy. Oncolytic vectors are engineered to replicate preferentially in cancer cells and to destroy cancer cells through the natural process of lytic virus replication. Many clinical trials indicate that replication-defective and replication-competent adenovirus vectors are safe and have therapeutic activity.

  1. Sections 2.2 is Replication deficient Ad and 2.3 is Modification Replication Deficient Ad. It is not clear what is the difference between the two groups of Ads. The authors stated that in line 175-176: “a "modification replication-deficient adenovirus" refers to an adenovirus that has been genetically altered to disableits ability to replicate efficiently within host cells”; in line 183-184: “A "replication-deficient adenovirus" refers to an adenovirus that has undergone genetic modifications to render itself incapable of replicating efficiently within host cells”. One is “disable” and another is “incapable”. Is there any difference between them.

Answer:

2.2

Engineered replication-defective adenoviruses (rAd) are widely used in vaccines and gene therapy due to their efficient expression of foreign inserts and minimized adenovirus protein expression. Notably, replication-deficient adenoviral (Ad) vaccines have been crucial during the SARS-CoV-2 pandemic, with several types receiving emergency use authorization and full approval in various regions. Despite some successful applications, such as the promising results of an RD Ad5-based tuberculosis vaccine, other trials like the STEP study and HVTN HIV vaccine trial faced setbacks due to unexpected outcomes and limited efficacy.

2.3

Replication-deficient adenoviruses are genetically modified to disable their ability to replicate efficiently within host cells, making them safer vectors for delivering therapeutic genes or antigens in gene therapy and vaccines. These modifications, which typically involve disabling or deleting key viral genes, prevent uncontrolled viral replication while allowing efficient gene delivery into target cells. This balance of safety and functionality makes replication-deficient adenoviruses valuable tools in various biomedical applications.

  1. In table 1, first and second generations of Ads are not for oncolytic therapies.

Answer: Corrected according to your suggestion.

  1. Line 615-623 is a well summary, but this does not belong to section 3.4. Vaccine Development.

Answer: It is the ending paragraph of section 3, that summarizes the overall views of the different approaches stated in the section 3.1 to 3.4. we think it’s a good concluding remarks of section 3.

  1. Minors that need check. Line 63-64: “which poses a major obstacle for the effectiveness of HAdV vectors [10] Similarly, administration of an AdV vector provokes…”

Answer: Line: 74-77, page: 02

which poses a significant obstacle to the effectiveness of HAdV vectors [10]. Similarly, the administration of an AdV vector provokes an anti-vector immune response, which can substantially hinder its effectiveness upon subsequent use.

Line 223-224: “They modified the Ad5 by adding a key component (ELDKWA) 223 known for broadly neutralizing various HIV-1 strains. Additionally, we included…”

Answer: Line: 254, page: 06

Additionally, they included the HIV- 1IIIB envelope (Env) gene in the vaccine to stimulate specific immune responses.

Round 2

Reviewer 2 Report

Comments and Suggestions for Authors

No further comments.